# Characterization of Thyroid Cancer among Hispanics in California, USA, from 2010 to 2020

**DOI:** 10.3390/cancers16061101

**Published:** 2024-03-08

**Authors:** Robert C. Hsu, Kai-Ya Tsai, David J. Benjamin, Krithika Chennapan, Katherine Y. Wojcik, Alice W. Lee, Jacob S. Thomas, Jorge J. Nieva, Lihua Liu

**Affiliations:** 1Department of Internal Medicine, Division of Medical Oncology, Keck School of Medicine, University of Southern California, Los Angeles, CA 90033, USA; krithika.chennapan@med.usc.edu (K.C.); jacob.thomas@med.usc.edu (J.S.T.); jorge.nieva@med.usc.edu (J.J.N.); lihualiu@usc.edu (L.L.); 2Norris Comprehensive Cancer Center, Keck School of Medicine, University of Southern California, Los Angeles, CA 90033, USA; 3Department of Population and Public Health Sciences, Keck School of Medicine, University of Southern California, Los Angeles, CA 90033, USA; kaiya.tsaierb@med.usc.edu (K.-Y.T.); kwojcik@usc.edu (K.Y.W.); 4Hoag Family Cancer Institute, Newport Beach, CA 92663, USA; david.benjamin@hoag.org; 5Department of Public Health, California State University, Fullerton, CA 92831, USA; alicelee@fullerton.edu

**Keywords:** thyroid cancer, risk factors, epidemiology, cancer disparities, Hispanics

## Abstract

**Simple Summary:**

Age-adjusted thyroid cancer incidence is lower in the Hispanic population than in the non-Hispanic White and Asian Pacific Islander populations. However, prior studies have shown an increased prevalence of advanced disease features such as larger tumor sizes and nodal involvement among Hispanics. We sought to characterize the demographic features and tumor characteristics of Hispanic thyroid cancer risk in California. We identified thyroid cancer cases from 2010 to 2020 using the California Cancer Registry. Overall, 56,638 diagnosed thyroid cancer cases were identified, including 16,852 (29.75%) Hispanics. Hispanics had the greatest female-to-male ratio disparity, average annual percentage change in incidence, and advanced disease features at diagnosis, as well as an increased mortality risk. After adjusting for demographic and tumor covariates, Hispanic ethnicity remained a significant independent variable for mortality risk. Consequently, further investigation into other possible factors associated with Hispanic ethnicity in thyroid cancer is needed.

**Abstract:**

Background: Previous studies on Hispanic thyroid cancer cases show sex disparities and an increased prevalence of large tumor sizes and nodal involvement. Here, we characterized Hispanic thyroid cancer cases in California. Methods: We identified thyroid cancer cases from 2010 to 2020 using the California Cancer Registry by sex, race/ethnicity, histology, TNM stage, tumor size, lymph node involvement, and Charlson comorbidity score. The age-adjusted incidence rate (AAIR) and age-adjusted mortality rate (AAMR) for all causes of death were calculated. A Cox proportional hazards regression analysis was performed to evaluate the mortality risk from all causes of death by race. Results: Overall, 56,838 thyroid cancer cases were identified, including 29.75% in Hispanics. Hispanics had the highest female-to-male incidence rate ratio (IRR 3.54) and the highest prevalence of T3/T4 tumor size (28.71%), the highest N1 nodal status (32.69%), and the highest AAMR (0.79 per 100,000 people). After adjusting for demographic and tumor covariates, compared to non-Hispanic White people, Hispanic ethnicity, with an HR of 1.22 (95% CI 1.18–1.25, *p* < 0.0001), remained a significant independent contributor to mortality risk. Conclusions: Hispanics had the greatest female-to-male IRR ratio, a greater prevalence of advanced disease features at diagnosis, along with the highest AAMR and increased mortality risk despite adjustments for demographic and tumor covariates. Further investigation into other risk factors is needed.

## 1. Introduction

The incidence of thyroid cancer has risen over much of the last 30 years, with Asian Pacific Islanders and non-Hispanic white people having the highest rates of incidence during this time [1]. Previous studies have attributed this increase in incidence among these ethnic groups to the screening of thyroid nodules and associated this with healthcare access [2].

While Hispanics have had a lower incidence rate of thyroid cancer compared to non-Hispanic White people and Asian Pacific Islanders, previous studies have shown that Hispanics have a higher percentage of metastatic disease upon presentation compared to non-Hispanic White people, along with a greater proportion of larger tumors and lymph node-positive disease at diagnosis, similar to that of Asian Pacific Islanders [3,4,5,6,7,8]. Within California, previous studies using California Cancer Registry (CCR) data from 1998 to 2004 have shown a higher female-to-male ratio of thyroid cancer incidence among Hispanics, along with a higher percentage of Hispanics being diagnosed younger than 55 years of age [9]. Another study from 1999 to 2008 of the CCR on patients with well-differentiated thyroid cancer showed a higher percentage of both metastatic and regional disease at diagnosis among Hispanics compared to non-Hispanic White people, consistent with other previous studies [6].

Furthermore, there have been mixed findings on the survival of Hispanic thyroid cancer patients. One study using CCR data from 1988 to 2010 found worse thyroid-cancer-specific survival among non-Hispanic Black and Hispanic adolescent young adult thyroid cancer patients [10]. In contrast, a Surveillance, Epidemiology, and End Results (SEER) registry analysis from 2010 to 2016 showed that Hispanics and Asian Pacific Islander thyroid cancer patients had significantly better survival than non-Hispanic White patients, while another SEER study from 2007 to 2011 showed that non-Hispanic Black patients had a worse 5-year mortality rate compared to non-Hispanic White patients, but no difference between Hispanic and non-Hispanic White patients was observed [3,11].

However, thyroid cancer incidence has continued to rise, and Hispanics now constitute the largest race/ethnicity in California as of the 2020 U.S. Census [12]. Thus, our study investigates whether these changes in incidence and changing demographics in California from a more recent time period have still correlated with the higher prevalence of advanced disease features in Hispanics and the associated mortality. In addition, we wanted to evaluate the sex disparity gaps suggested in previous studies between women and men. Finally, given the suggestions that thyroid cancer incidence is tied to overdiagnosis and access to healthcare, we evaluated socioeconomic status and healthcare insurance type to see if this plays a role in the incidence change rates and mortality risk between Hispanics and other races/ethnicities.

## 2. Methods

### 2.1. Data Source/Study Population

We used the California Cancer Registry (CCR) Research File of July 2023 to identify thyroid cancer cases diagnosed from 2010 to 2020 among California (CA) residents in a SEER*Stat readable format along with statewide mortality data with causes of death information. 

We examined demographic characteristics by sex (male, female), age at diagnosis (0–39, 40–49, 50–59, 60–69, 70+ years old), race/ethnicity (Hispanic, non-Hispanic (NH) White, NH Black, and Asian Pacific Islander (API)), and socioeconomic status (lowest, lower middle, middle, upper middle, highest). Socioeconomic status was derived from United States census data based on the patient’s address at the time of initial diagnosis while factoring in the following census variables using a principal components analysis: the proportion of people with a blue collar job, the proportion older than 16 years without a job, the median household income, the proportion of the population living below the 200% Federal Poverty Level, the median gross rent, the median value of owner-occupied houses, and the median education index [13,14]. Socioeconomic status was split into quartiles—lowest, lower middle, middle, upper middle, and highest.

The cases of thyroid cancer from 2010 to 2020 were identified using the International Classification of Disease for Oncology, Third Edition (ICD-O-3), with site code C73.9—thyroid gland. Tumor histotypes included follicular (ICD-O-3 codes 8290, 8330, 8331, 8332, 8333, 8335, 8337, 8339, 8346), papillary (ICD-O-3 codes 8050, 8260, 8261, 8262, 8263, 8340, 8341, 8342, 8343, 8344, 8347), medullary (ICD-O-3 codes 8345, 8510, 8346, 8347), and anaplastic (ICD-O-3 codes 8020, 8021, 8022).

We studied tumor behavior by evaluating the TNM stage (I-IV) along with the specific tumor size (T0–T4) and lymph node involvement (N0, N1) in cases from 2010 to 2020 using the American Joint Committee on Cancer (AJCC) staging 7th–8th editions [15,16]. We also looked at the Charlson comorbidity score from 2010 to 2020, categorizing them by 0 for no health comorbidities and 1+ for 1 or more health comorbidities.

The CCR collects limited treatment information within the first 6 months after a cancer diagnosis. Data specific to thyroid surgery (no surgery, lobectomy/local surgery, subtotal or near total thyroidectomy, total thyroidectomy, thyroidectomy/surgery, NOS, and unknown) were available from 2010 to 2020, along with radiation treatment data (no radiation, isotopes, radiation/combination/other, and unknown) from 2010 to 2020.

### 2.2. Statistical Analysis

To compare cancer risk levels among different groups, we calculated and presented the age-adjusted incidence rates and age-adjusted mortality rates by considering the number of cancer occurrences and cancer-related deaths, respectively, in relation to the size of a group’s at-risk population. The age-adjusted incidence rates (AAIRs) per 100,000 person-years of incidence and histology (papillary, follicular, medullary, anaplastic), along with the average annual percent change (AAPC) from 2010 to 2020, were calculated by sex and race/ethnicity using the CCR SEER*Stat database file of January 2023 [17]. In addition, we calculated the incidence rate ratio (IRR) of females to males. Additionally, age-adjusted mortality rates (AAMRs) for all causes of death were calculated by sex and race, with the AAPC measured from 2010 to 2020 using the California Department of Public Health’s Center for Health Statistics Death Master Files from 1970 to 2020 [18].

To evaluate the mortality risk from all causes of death by race/ethnicity in thyroid cancer cases, we performed a univariate analysis of race/ethnicity (Hispanic, NH White, NH Black, and API) and a multivariate hazard cox ratio regression analysis consisting of race/ethnicity (Hispanic, NH White, NH Black, and API), age (0–39, 40–64, 65+), sex (female, male), socioeconomic status (lowest, lower middle, middle, upper middle, highest), Charlson score (0, 1+), surgery (surgery, no surgery), radiation (isotopes, radiation/combination/other, no radiation), stage (I, II, III, IV), and lymphovascular invasion (present, not present). Given the mixed-race/ethnicity group, we excluded the American Indian/Other/Unknown race/ethnicity group from the univariate and multivariate analyses.

Tests for statistical significance were two-sided and considered statistically significant at *p* < 0.1. The statistical analysis was performed using SAS software, release 9.4. (SAS Institute, Cary, NC, USA).

## 3. Results

### 3.1. Demographics

Hispanics comprised 16,850 of the 56,638 total thyroid cancer cases from 2010 to 2020 in the state of California (Table 1). Hispanics had a higher percentage of cases with diagnoses at an age <40 years (34.42%) compared to the other races/ethnicities, with nearly 60% of Hispanic thyroid cancer cases being diagnosed before the age of 50. Hispanics had the highest female predominance (80.24%) among all the races/ethnicities (Table 1).

### 3.2. Disease Characteristics

As with all races/ethnicities, the papillary subtype was the most prevalent thyroid cancer subtype among Hispanics, accounting for 89.83%. In addition, 8.04% of thyroid cancer cases among Hispanics were Stage IV at diagnosis, and when considering specific tumor sizes and nodal staging, Hispanics had the highest percentage of the T3 (22.40%) and T4 tumor stages (6.31%), along with the N1 nodal stage (32.69%). Furthermore, 23.04% of Hispanic thyroid cancer patients were found to have a Charlson comorbidity score of one or higher (Table 2).

### 3.3. Treatment

In terms of treatment, 77.42% of Hispanic thyroid cancer patients underwent a total thyroidectomy, which was the highest proportion among all the races/ethnicities. In addition, 29.99% of Hispanic thyroid cancer patients had radioactive iodine therapy, and 1.85% received a combination of a thyroidectomy and radiation therapy (Table 3).

### 3.4. AAIR 

The overall AAIR of thyroid cancer among Hispanics from 2010 to 2020 was the third highest amongst all the races, at 12.11 cases per 100,000 person years, with an AAIR of 5.31 per 100,000 person years in males and 18.81 per 100,000 person years in females. Hispanics had the highest AAPC of 1.0 (95% CI −0.3–2.4, *p* < 0.1) overall and also when stratified by males, with an AAPC of 3.4 (95% CI 1.4–5.5, *p* < 0.1), and females, with an AAPC of 0.6 (95% CI −1.2–2.5, *p* < 0.1). Hispanics had the highest female-to-male incidence rate ratio (IRR) of 3.54 compared to the other races/ethnicities (Figure 1 and Figure 2; Table 4 and Table 5).

### 3.5. AAMR

Hispanics had the highest AAMR from 2010 to 2020 at 0.79 per 100,000 person-years with an AAPC of 1.6 (95% CI −3.2–6.6, *p =* 1). Stratified by sex, Hispanic males had the highest AAMR from 2010 to 2020 of 0.62 per 100,000 person-years, with an AAPC of 4.2 (95% CI −6.9–16.6, *p* < 0.1), and Hispanic females also had the highest AAMR from 1988 to 2020 of 0.86, with an AAPC of 0.9 (95% CI −2.3–4.3, *p =* 1). Hispanics had the second highest female-to-male IRR of 1.38 (Figure 3; Table 6 and Table 7).

### 3.6. Adjusted Cox Hazard Ratios

The univariate analysis demonstrated that Hispanic ethnicity had the highest risk of mortality among all the races/ethnicities with an HR of 1.22 (95% CI 1.19–1.24, *p* < 0.0001) in both females, with an HR of 1.24 (95% CI 1.21–1.27, *p* < 0.0001), and males, with an HR of 1.19 (95% CI 1.14–1.24, *p* < 0.0001) (Table 8).

When addressing disease characteristics and socioeconomic variables (race/ethnicity, sex, age, socioeconomic status, Charlson comorbidity score, AJCC stage, lymphovascular invasion, and treatment), Hispanic ethnicity had a significant mortality risk, with an overall HR of 1.21 (95% CI 1.18–1.24, *p* < 0.0001), in both females, with an HR of 1.21 (95% CI 1.17–1.24, *p* < 0.0001), and males, with an HR of 1.20 (95% 1.14–1.27, *p* < 0.0001). Other significant risk factors included Asian Pacific Islander race (overall HR 1.12, 95% CI 1.09–1.15, *p* < 0.0001), age 65+ (overall HR 1.14, 95% CI 1.11–1.18, *p* < 0.0001), male sex (overall HR 1.12, 95% CI 1.09–1.15, *p* < 0.0001), middle (overall HR 1.06, 95% CI 1.03–1.10), lower middle (overall HR 1.05, 95% CI 1.01–1.09, *p* = 0.0007), and lowest socioeconomic status (overall HR 1.09, 95% CI 1.04–1.13, *p* < 0.0001), and stage II (overall HR 1.17, 95% CI 1.13–1.22, *p* < 0.0001) and stage IV thyroid cancer (overall HR 1.09, 95% CI 1.04–1.15, *p* < 0.0001) (Table 9).

## 4. Discussion

In our analysis of Hispanic thyroid cancer patients in California from 2010 to 2020, we show that the more than 3.5:1 ratio of female to male was the largest compared to all the other races/ethnicities, and that Hispanics had the greatest AAIR AAPC, with Hispanic males (3.4; 95% CI 1.4–5.5, *p* < 0.1) having a significantly positive AAIR AAPC that no other male or female racial/ethnic group demonstrated. Furthermore, a greater proportion of Hispanics were diagnosed with advanced disease features, including larger tumor sizes, positive nodal sizes, and lymphovascular invasion. Hispanic ethnicity had the highest AAMR among both males and females, with Hispanic males having a significantly increased AAPC (4.2; 95% CI −6.9–16.6, *p* < 0.1). This finding persisted after adjusting for different variables, including both socioeconomic and tumor-related variables; Hispanic ethnicity in California (HR: 1.21, 95% CI 1.18–1.24, *p* < 0.0001) persistently remained a significantly independent variable in mortality risk in thyroid cancer. Thus, our findings are consistent with studies from past years but also show a significant mortality risk among Hispanics.

In thyroid cancer, estrogen receptor-alpha cells are overexpressed, while estrogen receptor-beta cells reduce tumor growth, and the fact that the ratio of estrogen receptor-alpha is greater than estrogen receptor-beta in thyroid cancer contributes to the high female-to-male ratio. Hispanics have been shown to have similar estrogen levels to non-Hispanic White people but lower levels than non-Hispanic Black people [19,20]. But, obesity appears to be associated with the development of differentiated thyroid cancer, particularly with leptin, a hormone produced primarily by adipose tissue; leptin and its receptor (OB-R) have been implicated to be overexpressed in thyroid carcinomas, particularly papillary thyroid cancer, as multiple studies of PTC tissues have shown an expression of 50–80% of PTC tissue expressing OB-R and leptin [21,22,23,24]. Hispanics (44.8%) do have higher rates of obesity compared to non-Hispanic White people (42.2%) and Asian Pacific Islanders (17.4%), but similar rates of obesity exist between Hispanic males (45.7%) and females (43.7%) [25]. Much of the sex disparity lies in the fact that many of the cases among women were small, early-stage papillary thyroid cancer cases [26]. Yet, in our data, we see that Hispanics have a greater prevalence of large tumor sizes. We note that thyroid cancer is the second most common cancer diagnosis during pregnancy, and there have been links to recent pregnancy and infertility problems causing thyroid cancer [27,28,29]. Compared to Non-Hispanic women, Hispanic women with Medicaid insurance had 22% higher rates of severe pregnancy complications in 2020 and Hispanic women with private insurance had a 28% increase in severe pregnancy complications in 2021 [30]. The sex ratio in thyroid cancer may also be genetic, as six single-nucleotide polymorphisms (SNPs) identified in Colombia have been tied to an increased female-to-male ratio (5:1) along with larger tumor sizes [31]. As such, further investigations are needed to understand the greater sex disparity in thyroid cancer incidence among Hispanics.

Hispanics were also found to have the highest AAIR AAPC (1.0; 95% CI −0.3–2.4, *p* < 0.1), with Hispanic males having a significantly positive AAIR AAPC (3.4; 95% CI 1.4–5.5, *p* < 0.1) that no other male or female racial group demonstrated. This could be multifactorial; first, Hispanic presence in California has been increasing, as the percentage of Hispanics has increased from 19.8% in 1980 to 32.4% in 2000 to 37.6% in 2010 to 40.4% in 2020 [12,32]. Furthermore, as mentioned prior, Hispanics have higher percentages of obesity than non-Hispanic White people, which may be contributing to the AAPC [25]. In addition, it has been suggested that metabolic syndrome and its subsequent abnormal metabolism can place the body in a state of chronic inflammation, leading to thyroid-stimulating hormones inducing tumorigenesis [33]. Hispanics have been shown to have a higher prevalence of cardiovascular disease, Type II diabetes, non-alcoholic fatty liver disease, and dyslipidemia, which may indirectly play into the higher increases in thyroid cancer incidence [34].

Our data also demonstrated that Hispanics had a higher prevalence of larger tumor sizes and positive nodal sizes compared to other races/ethnicities, consistent with previous studies [3,4,5,6,7,8]. An SEER analysis from 2017 to 2018 showed that Hispanics had a greater percentage of nodal disease at initial surgery compared to non-Hispanic White people and a greater incidence of recurrent disease, while another SEER analysis showed that Hispanics were more likely to present with distant disease than non-Hispanic White people [5]. Meanwhile, a meta-analysis has shown that Hispanics have the greatest proportion among all races/ethnicities of having tumors > 4 cm in size at diagnosis [4]. Finally, another SEER analysis from 1992 to 2006 showed that Hispanics had the highest rate of anaplastic thyroid cancer diagnosed [35]. This was also seen among papillary thyroid cancer patients in an SEER analysis from 2011 to 2013 receiving adjuvant radiation, in which Hispanics and Asian Pacific Islanders had greater odds of T4, N1, or M1 disease [36].

Socioeconomic status and access to care may factor into the increased mortality risk of Hispanic ethnicity, as middle-to-low socioeconomic status was a significant independent variable in our multivariate analysis. A total of 47% of Hispanic women in Los Angeles with thyroid cancer reported financial hardship, along with 49% reporting low acculturation to the English language [37]. While Hispanics have a lower AAIR of thyroid cancer compared to Asian Pacific Islanders and non-Hispanic White people, uninsured Hispanics had the highest AAIR compared to the other races/ethnicities [1]. Hispanics have the lowest health insurance rate of any racial or ethnic group (34%), including more than twice that of non-Hispanic White people [38,39]. Hispanics also have a 25% lower median income than non-Hispanic White people [39,40]. Subsequently, the limitations of financial resources and access to care may affect Hispanics in terms of having adequate health literacy and limit their means to being able to have a well-balanced diet and places to carry out ample exercise, which are key factors in avoiding some possible triggers to thyroid cancer such as metabolic syndrome and obesity.

Despite factoring disease characteristics and socioeconomic variables into our multivariate analysis, Hispanic ethnicity (HR: 1.21, 95% CI 1.18–1.24, *p* < 0.05) remains a significant risk factor. Some of this may have to do with genetic changes in Hispanics relative to other races/ethnicities. Hispanics have been shown to have an increased prevalence of *BRAF* V600E, and this mutation has been linked in the past with increased mortality [41,42]. The *BRAF* V600E mutation has been shown to be associated with more advanced clinical stages and decreased responsiveness to radioiodine [43]. Even in smaller-size papillary thyroid cancer cases of <1.5 cm, a significantly higher percentage of *BRAF* V600E cases demonstrate aggressive features such as extrathyroidal extension and lymph node metastasis compared to non-mutated cases [44]. Among Hispanics, poorly differentiated *BRAF* mutant cases are significantly increased compared to non-Hispanic White people [43,45]. Furthermore, a Colombian study investigating 141 papillary thyroid cancer cases showed that the *BRAF* V600E and *TERT* C228T or C250T mutations were associated with large tumors, lymph node metastasis, extra-thyroid extension, and advanced stages, and compared to published data on U.S. white people, Colombian patients had a higher prevalence of both severe pathologic features and double-mutant tumors [46]. It has been thought that some of the key genes triggering this are related to several microRNAs targeting the TGF-beta–SMAD pathway [43]. *TERT* promoter mutations have been shown in 95% of thyroid tumors that have transformed from papillary carcinoma to a more aggressive anaplastic thyroid carcinoma and so pose a greater risk of transformation to a more aggressive and rare thyroid cancer compared to other mutations [47]. That being said, additional studies with larger sample sizes comparing *BRAF* V600E-mutant thyroid cancer cases among Hispanics and non-Hispanics are merited to better characterize the effect of *BRAF* V600E on the prevalence of advanced features and mortality risk. 

There are clinical implications, as dabrafenib, a *BRAF*-mutated tyrosine kinase inhibitor, has been shown to have efficacy in well-differentiated radioactive-iodine-resistant thyroid cancer patients and, in combination with trametinib, has been shown to improve overall survival in anaplastic thyroid cancer [48,49]. Other important mutations to note that it could be worth investigating the prevalence of among Hispanics given their associations with anaplastic transformation outside of *TERT*-promoter mutations include subunits of the SWI/SNF complex such as *SMARCA*4 and alterations of the *PIK3CA* gene, which were found to be positive in 33% of thyroid tumor samples with a heterogenous pathology of papillary and anaplastic carcinoma components [47]. Mutations in *RAS* genes occur in 30–45% of follicular thyroid cancer, 30–45% of follicular-variant papillary thyroid cancer, and 20–40% of poorly differentiated thyroid cancer and are also worth investigation among Hispanics [50]. Meanwhile, *RET* proto-oncogene is found in medullary thyroid carcinoma, and while a small-case analysis of medullary thyroid carcinoma among Mexicans has been performed, larger studies could be conducted to evaluate any possible connection between any particularly Hispanic subgroups and medullary thyroid cancer/multiple endocrine neoplasia type 2 [51,52].

By studying additional molecular alterations in Hispanics relative to other races/ethnicities, this may help mitigate the mortality risk among the Hispanic race/ethnicity. With possible known mutations and associated targeted therapies, the need for equality in obtaining comprehensive genomic profiling of tumors and access to targeted therapy agents is critical. Hispanic ethnicity has been shown to be underrepresented among patients obtaining precision oncology assays [53]. Moreover, Hispanics have had less access to targeted drugs and are severely underrepresented in clinical trials that could provide access to these drugs and help better understand biological differences in response to these drugs [54].

The key limitations of this analysis and important considerations moving forward are that our analysis consists of Hispanics in the state of California in the United States, which means that it is not completely generalizable to Hispanic populations elsewhere. Hispanics in California consist of primarily those with Mexican descent (77%), while Hispanics in Florida consist of those with primarily Caribbean descent, notably Cuban (28%) and Puerto Rican (21%) [55,56]. Further analysis in other sectors of the U.S. may help better elucidate differences in tumor characteristics and socioeconomic challenges among different Hispanic subgroups. In addition, another limitation is the lack of genetic testing available. Comparing molecular testing for key mutations of thyroid cancer, such as *BRAF* V600E and *RET*, is important among Hispanics and other races/ethnicities and within Hispanic subgroups. 

Another important aspect that is critical in future analysis is comparing differences in thyroid cancer among Hispanic immigrants from different origins and Hispanics who are U.S.-born. U.S.-born Hispanic women younger than 55 have been shown to have a significantly greater incidence of papillary thyroid cancer, while the opposite is true in older Hispanic women [9]. In another study comparing Colombians and U.S. Hispanics, U.S.-born Hispanics had a smaller mean tumor size of 20 mm vs. 27 mm than Colombians and were half as likely to have regional disease (16.7% vs. 31.15%) [31]. In another study comparing Puerto Ricans and U.S. mainland Hispanics, Puerto Rican women had an 83% increased risk of a papillary thyroid cancer diagnosis compared to U.S. Hispanics, and Puerto Rican men had a 2.2-fold increased risk compared to U.S. Hispanic men [57]. This shows that disaggregated data among Hispanics both in the U.S. and outside the U.S. by origin will be important in helping describe mortality risk in thyroid cancer. Finally, 5% of the population in California is made up of undocumented immigrants, with Mexicans and Salvadorans constituting the largest share [58]. Given that some of their data may not be identifiable in the cancer registry as a result, the magnitude of the risks involved in the incidence and mortality of thyroid cancer may be even greater.

Thus, our study highlights key important factors showing that greater attention needs to be paid to Hispanics in thyroid cancer moving forward due to the greater prevalence of large tumor sizes and positive nodal status at presentation, the greater sex disparities in women compared to men, and the increased mortality driven both by tumor characteristics and socioeconomic factors among Hispanics. As much of this could be tied to differences in the genetic makeup of thyroid cancer tumors, access to molecular testing, targeted therapies, clinical trials, and mitigating financial costs is paramount to helping bridge these disparities. 

## 5. Conclusions

The thyroid cancer incidence among Hispanics trails behind Asian Pacific Islanders and non-Hispanic White people, but previous studies have shown that the presentation of thyroid cancer cases among Hispanics has higher incidences of large tumor sizes, nodal involvement, and distant metastasis. There also appear to be possible financial barriers and concerns about undertreatment leading to worse outcomes in thyroid cancer among Hispanics. In our analysis using the California Cancer Registry from 2010 to 2020, we demonstrated that Hispanics have the largest sex disparities, as Hispanic women have an incidence of 3.5-fold that of Hispanic men. While there is no clear reasoning for this, possible risk factors include obesity, infertility, and genetic alterations. Further, we showed that Hispanics had the highest AAPC, which could be tied into not only demographics but other factors such as high obesity rates. Consistent with previous studies, our analysis demonstrated that Hispanics had the largest proportion of T3/T4 tumor sizes and nodal involvement. Ultimately, in our multivariate analysis for thyroid cancer mortality, which included both tumor characteristics and socioeconomic factors, Hispanic race/ethnicity remained a significant risk factor. While we believe that this is multifactorial secondary to both socioeconomic factors and the higher prevalence of high-risk disease features such as large tumor sizes and nodal involvement, the incidence of genetic alterations of mutations such as *BRAF* V600E and *TERT* could play a role in such findings. Further investigation to better understand the sex disparities seen and the increased mortality risk among Hispanic thyroid cancer cases will require more analysis of genetic alterations among the Hispanic population, a disaggregation of data within Hispanic subgroups with thyroid cancer, and a study of the differences in thyroid cancer between U.S.-born and foreign-born Hispanic patients.

## Figures and Tables

**Figure 1 cancers-16-01101-f001:**
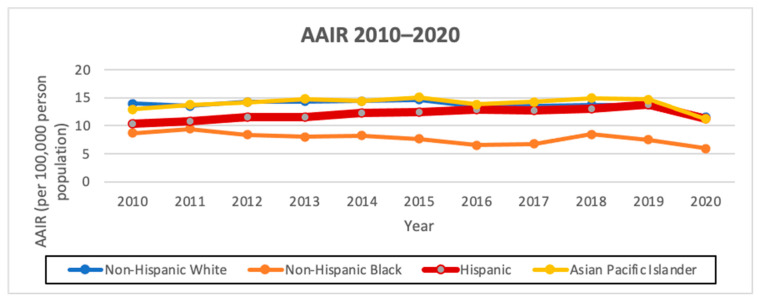
AAIR of overall thyroid cancer incidence from 2010 to 2020 in California by race/ethnicity (per 100,000 people).

**Figure 2 cancers-16-01101-f002:**
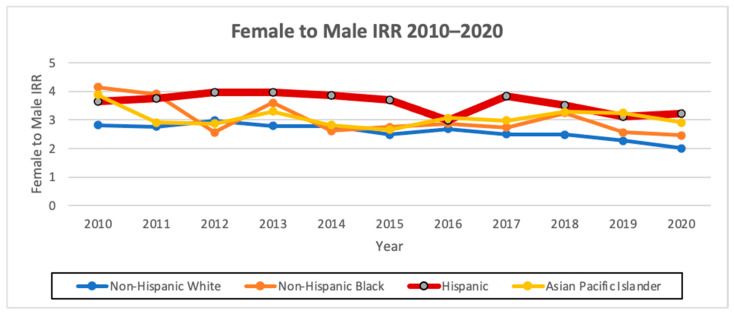
Female-to-male AAIR ratio from 2010 to 2020 in California by race/ethnicity.

**Figure 3 cancers-16-01101-f003:**
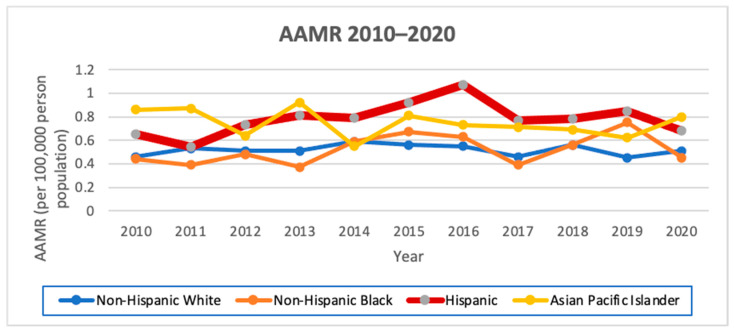
AAMR from 2010 to 2020 in California by race/ethnicity.

**Table 1 cancers-16-01101-t001:** Demographics of thyroid cancer cases in California from 2010 to 2020.

	Non-Hispanic White	Non-Hispanic Black	Hispanic	Asian Pacific Islander	American Indian/Other/Unknown	Total
Sex (Missing 8 cases) (%)						
Male	7874 (29.42)	491 (23.64)	3328 (19.75)	2136 (21.79)	269 (23.56)	14,098
Female	18,885 (70.56)	1585 (76.31)	13,522 (80.24)	7667 (78.20)	873 (76.44)	42,532
Total	26,759	2076	16,850	9803	1142	56,630
Age at Diagnosis (%)						
<40 years	6010 (22.46)	404 (19.45)	5801 (34.42)	2569 (26.20)	373 (32.66)	15,157
40–49 years	4715 (17.62)	405 (19.50)	3811 (22.61)	2133 (21.76)	244 (21.37)	11,308
50–59 years	6302 (23.55)	531 (25.57)	3493 (20.73)	2085 (21.27)	273 (23.91)	12,684
60–69 years	5529 (20.66)	420 (20.22)	2260 (13.41)	1803 (18.39)	156 (13.66)	10,168
70+ years	4207 (15.72)	317 (15.26)	1487 (8.82)	1214 (12.38)	96 (8.41)	7321
Total	26,763	2077	16,852	9804	1142	56,638

**Table 2 cancers-16-01101-t002:** Disease characteristics of thyroid cancer cases in California from 2010 to 2020 by race/ethnicity. “-” A case count smaller than 11 is withheld to protect patient confidentiality and the stability of the statistical estimation.

	Non-Hispanic White	Non-Hispanic Black	Hispanic	Asian Pacific Islander	American Indian/Other/Unknown	Total
Tumor Histotype (%)						
Papillary	23,407 (87.46)	1681 (80.93)	15,138 (89.83)	8917 (90.95)	1013 (88.70)	50,156
Follicular	1900 (7.10)	248 (11.94)	949 (5.63)	499 (5.09)	74 (6.48)	3670
Medullary	549 (2.05)	41 (1.97)	281 (1.67)	93 (0.95)	15 (1.31)	979
Anaplastic	290 (1.08)	-	163 (0.97)	104 (1.06)	-	586
Missing	617 (2.31)	-	321 (1.90)	191 (1.95)	-	1247
Total	26,763	2077	16,852	9804	1142	56,638
TNM Stage (%)						
I	17,490 (65.35)	1305 (62.83)	11,747 (69.71)	6399 (65.27)	776 (67.95)	37,717
II	2603 (9.73)	235 (11.31)	1171 (6.95)	788 (8.04)	58 (5.08)	4855
III	2841 (10.62)	186 (8.96)	1596 (9.47)	1126 (11.49)	72 (6.30)	5821
IV	2210 (8.26)	175 (8.43)	1355 (8.04)	897 (9.15)	39 (3.42)	4676
Missing	1619 (6.05)	176 (8.47)	983 (5.83)	594 (6.06)	197 (17.25)	3569
Total	26,763	2077	16,852	9804	1142	56,638
Tumor Stage (%)						
T0/T1	14,976 (55.96)	1049 (50.51)	8075 (47.92)	4953 (50.52)	522 (45.71)	29,575
T2	4628 (17.29)	379 (18.25)	2991 (17.75)	1629 (16.62)	166 (14.54)	9793
T3	4703 (17.57)	398 (19.16)	3775 (22.40)	2157 (22.00)	162 (14.19)	11,195
T4	1176 (4.39)	98 (4.72)	1064 (6.31)	555 (5.66)	27 (2.36)	2920
Tx	955 (3.57)	116 (5.58)	729 (4.33)	402 (4.10)	200 (17.51)	2402
NA	192 (0.72)	-	101 (0.60)	52 (0.53)	-	396
Missing	133 (0.50)	-	117 (0.69)	56 (0.57)	-	357
Total	26,763	2077	16,852	9804	1142	56,638
Nodal Stage (%)						
N0	18,426 (68.85)	1564 (75.30)	10,157 (60.27)	6161 (62.84)	625 (54.73)	36,933
N1	6756 (25.24)	303 (14.59)	5509 (32.69)	3019 (30.79)	220 (19.26)	15,807
NA	185 (0.69)	-	100 (0.59)	52 (0.53)	-	388
Unknown	1263 (4.72)	173 (8.33)	969 (5.75)	517 (5.27)	232 (20.32)	3154
Missing	133 (0.50)	-	117 (0.69)	55 (0.56)	-	356
Total	38,386	2077	16,852	9804	1142	56,638
Charlson Score (%)						
0	14,991 (56.01)	898 (43.24)	8210 (48.72)	4932 (50.31)	448 (39.23)	29,479
1+	6135 (22.92)	752 (36.21)	3882 (23.04)	1963 (20.02)	200 (17.51)	12,932
Missing	5637 (21.06)	427 (20.56)	4760 (28.25)	2909 (29.67)	494 (43.26)	14,227
Total	26,763	2077	16,852	9804	1142	56,638

**Table 3 cancers-16-01101-t003:** Surgery/radiation by race/ethnicity. “-” A case count smaller than 11 is withheld to protect patient confidentiality and the stability of the statistical estimation.

Surgery (%)	Non-Hispanic White	Non-Hispanic Black	Hispanic	Asian Pacific Islander	American Indian/Other/Unknown	Total
No surgery	1595 (5.96)	154 (7.41)	1068 (6.34)	706 (7.20)	250 (21.89)	3773
Lobectomy/local surgery	2752 (10.28)	260 (12.52)	1446 (8.58)	1086 (11.08)	135 (11.82)	5679
Subtotal or near total thyroidectomy	400 (1.49)	50 (2.41)	309 (1.83)	136 (1.39)	16 (1.40)	911
Total thyroidectomy	20,239 (75.62)	1459 (70.25)	13,046 (77.42)	7309 (74.55)	656 (57.44)	42,709
Thyroidectomy/surgery, NOS	191 (0.71)	-	153 (0.91)	69 (0.70)	-	448
Unknown	41 (0.15)	-	23 (0.14)	11 (0.11)	-	85
Missing	2167 (5.42)	126 (6.07)	807 (4.79)	487 (4.97)	68 (5.95)	3033
Total	26,763	2077	16,852	9804	1142	56,638
Isotopes	7688 (28.73)	507 (24.41)	5054 (29.99)	2943 (30.02)	205 (21.89)	16,397
No radiation	18,523 (69.21)	1532 (73.76)	11,460 (68.00)	6881 (68.15)	920 (80.56)	39,116
Radiation/combination/other	482 (1.80)	31 (1.49)	311 (1.85)	168 (1.71)	14 (1.23)	1006
Unknown/missing	70 (0.26)	-	27 (0.16)	12 (0.12)	-	119
Total	26,763	2077	16,852	17,267	1142	56,638

**Table 4 cancers-16-01101-t004:** AAIR and IRR from 2010 to 2020 in California by race/ethnicity/sex (per 100,000 people).

Race/Ethnicity	Total AAIR from 2010 to 2020	Total AAIR in 2020	Male AAIR from 2010 to 2020	Male AAIR in 2020	Female AAIR from 2010 to 2020	Female AAIR in 2020	IRR Female to Male 2010 to 2020
NH White	13.76	11.61	7.72	7.8	20.02	15.63	2.59
Hispanic	12.11	11.19	5.31	5.29	18.81	17.07	3.54
API	13.98	11.20	6.65	5.56	20.34	16.16	3.06
NH Black	7.75	5.98	3.87	3.49	11.47	8.58	2.96

**Table 5 cancers-16-01101-t005:** AAIR AAPC in California by race/ethnicity/sex from 2010 to 2020.

Race/Ethnicity	Total AAPC from 2010 to 2020 (95% CI)	Female AAPC from 2010 to 2020 (95% CI)	Male AAPC from 2010 to 2020 (95% CI)
NH White	−1.0 (−2.2–0.2)	−2.6 (−4.8−0.3)	1.4 (0.1–2.2)
Hispanic	1.0 (−0.3–2.4)	0.6 (−1.2–2.5)	3.4 (1.4–5.5)
API	−1.4 (−4.0–1.2)	−1.4 (−3.5–0.7)	1.7 (−3.7–7.4)
NH Black	−3.0 (−5.1–−0.9)	−3.6 (−5.9–−1.2)	0.1 (−2.7–2.9)

**Table 6 cancers-16-01101-t006:** AAMR and IRR overall from 2010 to 2020 in California by race/ethnicity/sex (per 100,000 people).

Race/Ethnicity	Total AAMR from 2010 to 2020	Total AAMR in 2020	Male AAMR from 2010 to 2020	Male AAMR in 2020	Female AAMR in 2010 to 2020	Female AAMR in 2020	IRR Female to Male 2010 to 2020
NH White	0.52	0.51	0.55	0.69	0.49	0.37	0.89
Hispanic	0.79	0.68	0.62	0.68	0.86	0.76	1.38
API	0.74	0.80	0.57	0.61	0.83	0.86	1.51
NH Black	0.52	0.45	0.45	0.43	0.57	0.47	1.27

**Table 7 cancers-16-01101-t007:** AAMR by race/ethnicity (per 100,000 people) by year from 2010 to 2020.

Race/Ethnicity	Total AAPC from 2010 to 2020 (95% CI)	Female AAPC from 2010 to 2020 (95% CI)	Male AAPC from 2010 to 2020 (95% CI)
NH White	−0.1 (−2.2- 1.9)	−2.3 (−5.2–0.7)	1.9 (−1.1–5.1)
Hispanic	1.6 (−3.2–6.6)	0.9 (−2.3–4.3)	4.2 (−6.9–16.6)
API	−1.6 (−4.9–1.9)	−2.0 (−5.3–1.5)	1.7 (−3.7–7.4)
NH Black	3.1 (−1.9–8.4)	2.8 (−6.2–12.6)	11.0 (−1.9–25.6)

**Table 8 cancers-16-01101-t008:** Univariate hazard ratio Cox regression analysis 2010–2020.

		HR	HR (95% CI)		HR	HR (95% CI)		HR	HR (95% CI)	
		n = 55,349 (Total)			n = 41,579 (Female)			n = 13,770 (Male)		
Race	Non-Hispanic White	Reference			Reference			Reference		
	Non-Hispanic Black	1.02	0.97	1.07	1.01	0.96	1.06	1.10	1.00	1.20
	Hispanic	1.22	1.19	1.24	1.24	1.21	1.27	1.19	1.14	1.24
	Asian/Pacific Islander	1.12	1.17	1.21	1.14	1.11	1.17	1.08	1.03	1.13

**Table 9 cancers-16-01101-t009:** Multivariate hazard ratio Cox regression analysis with race, age, sex, socioeconomic status, treatment, Charlson score, stage, and lymphovascular invasion data from 2010 to 2020 (excluding missing total n: sex: 8 (0.01%) missing, Charlson score: 13,692 (24.73%) missing, stage: 3284 (5.93%) missing, radiation: 56 (0.10%) unknown/missing, and surgery: 2987 (5.40%) unknown/missing; missing female n: Charlson score: 10,326 (24.83%) missing, stage: 2373 (5.71%) missing, radiation: 47 (0.11%) unknown/missing, and surgery: 2260 (5.44%) unknown/missing; and missing male n: Charlson score: 3361 (24.41%) missing, stage: 911 (6.62%) missing, radiation: 9 (0.07%) unknown/missing, and surgery: 721 (5.24%) unknown/missing).

		HR	HR (95% CI)		HR	HR (95% CI)		HR	HR (95% CI)	
		n = 37,395			n = 28,110 (Female)			n = 9285 (Male)		
Race	Non-Hispanic White	Reference			Reference			Reference		
	Non-Hispanic Black	0.97	0.91	1.02	0.94	0.89	1.00	1.07	0.95	1.20
	Hispanic	1.21	1.18	1.24	1.21	1.17	1.24	1.20	1.14	1.27
	Asian/Pacific Islander	1.12	1.09	1.15	1.13	1.10	1.17	1.06	1.00	1.12
Age	0–39	Reference			Reference			Reference		
	40–64	0.95	0.93	0.98	0.95	0.92	0.98	0.96	0.90	1.02
	65+	1.14	1.11	1.18	1.12	1.08	1.16	1.19	1.12	1.28
Sex	Female	Reference								
	Male	1.12	1.09	1.15						
Socioeconomic Status	Highest	Reference			Reference			Reference		
	Upper middle	1.01	0.98	1.04	1.01	0.97	1.04	1.02	0.96	1.08
	Middle	1.06	1.03	1.10	1.06	1.02	1.10	1.08	1.01	1.14
	Lower middle	1.06	1.02	1.09	1.05	1.01	1.09	1.08	1.01	1.15
	Lowest	1.08	1.04	1.12	1.09	1.04	1.14	1.03	0.96	1.11
Charlson Score	1+	Reference			Reference			Reference		
	0	0.85	0.83	0.87	0.86	0.84	0.89	0.82	0.78	0.86
Surgery	Surgery	Reference			Reference			Reference		
	No surgery	2.20	2.09	2.32	2.17	2.03	2.32	2.21	2.02	2.42
Radiation	No radiation	Reference			Reference			Reference		
	Isotopes	0.57	0.56	0.58	0.57	0.56	0.59	0.56	0.54	0.59
	Radiation/combination/other	0.87	0.80	0.93	0.82	0.75	0.90	0.94	0.84	1.05
Stage	I	Reference			Reference			Reference		
	II	1.21	1.16	1.25	1.20	1.15	1.26	1.21	1.13	1.29
	III	0.99	0.96	1.03	1.01	0.97	1.05	0.94	0.88	1.00
	IV	1.34	1.28	1.39	1.39	1.32	1.46	1.25	1.18	1.33

## Data Availability

The data presented in this study are available in this article.

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
