# Peer review of "Characterization of Thyroid Cancer among Hispanics in California, USA, from 2010 to 2020"

_cancers, 2024, doi:10.3390/cancers16061101_

Round 1
Reviewer 1 Report
Comments and Suggestions for Authors
The authors used retrospective data to analyse differences in thyroid cancer incidence, mortality and characteristics between ethnicities living in California. The study involves a detailed statistical analysis and considers a multitude of factors as putative causes in observed discrepancies between ethnicities. However, the paper resembles a demographic study rather than a research aiming to elucidate the causes of differences in cancer incidence. The discussion section is baffling, and often widely comments on facts that are not directly related to the research. Also, till the end, it is difficult to understand whether it is the biological or socioeconomical factors that contribute to divergent susceptibility to thyroid cancer (or its clinicopathological factors). That said, the manner in which the authors estimated the socioeconomical status of subjects is very questionable. In my opinion, the manuscript is not satisfactory to be published in Cancers
Author Response
Thank you for sharing your thoughts on this. This paper is an analysis of data from a cancer registry and so demographics will be a part of the analysis as acknowledged. However, there are key factors that go beyond basic demographics such as stage and treatment history. The discussion is meant to take these conclusions and consider possible reasons; we did work to help soften some of the conclusions in the discussion to make them more as possibilities to consider instead of causative such as the discussion on BRAF V600E mutations in Hispanic thyroid cancer patients. Unfortunately, one of the biggest issues with cancer research has been the lack of studies involving Hispanics particularly from a biological aspect; Hispanics are very underrepresented in clinical trials and little information is out there on some of the genomics that are probably driving these changes but don’t exist and as a result limit how we can describe these findings. Thus, by trying to connect possibilities driving the clear conclusion that Hispanics have greater age-adjusted incidence and mortality risk in Hispanic thyroid cancer patients in California, this will motivate scientists to do further studies that are more inclusive to Hispanics that will then help build on these conclusions that we find in the California Cancer Registry.

Reviewer 2 Report
Comments and Suggestions for Authors
Reviewer comments on manuscript # cancers-2868735, entitled “Characterization of Thyroid Cancer among Hispanics in California, USA from 2010-2020”,
The manuscript is generally well written and identifies, by ethnicity, a higher age adjusted percentage change in the incidence of thyroid cancer in Hispanics in California in the second decade of the 21th century. They also found a marginally higher incidence of age adjusted mortality ratio due to thyroid cancer in Hispanics (0.79) when compared with non-hispanic whites/blacks and Asian/pacific islanders (0.52 to 0.74).
Comment #1, all manuscript: there are several editing errors where words are separated by a hyphen.
Comment #2, lines 34 and 125: mortality NOT morality.
Comment #3, methods, results and discussion sections: the authors state in the methods section that statistical significance was considered if p<0.05, and use the word significantly in the discussion section to state differences in several outcomes, namely age adjusted mortality ratio and age adjusted percentage change in incidence. However, they do not provide p values in the results, neither in the text nor in tables/figures. Thus, p values should be shown (in the text or in the tables/figures) to reinforce their message.
Comments on the Quality of English LanguageThe manuscript is generally well written.
Author Response
Comment #1, all manuscript: there are several editing errors where words are separated by a hyphen.
Response: Noted, we have reviewed the manuscript and removed these hyphens where thought to be in words that should not be separated by a hyphen.
Comment #2, lines 34 and 125: mortality NOT morality.
Response: Noted, we have made these edits.
Comment #3, methods, results and discussion sections: the authors state in the methods section that statistical significance was considered if p<0.05, and use the word significantly in the discussion section to state differences in several outcomes, namely age adjusted mortality ratio and age adjusted percentage change in incidence. However, they do not provide p values in the results, neither in the text nor in tables/figures. Thus, p values should be shown (in the text or in the tables/figures) to reinforce their message.
Response: Noted, we have added the p-values where appropriate into the text of the results and discussion. The significance of the p-value was adjusted from p<0.05 to p<0.1 because for our AAIR and AAMR AAPC statistical testing the SAS output would read the p-value as <0.1. In our univariate and multivariate analysis, the p-value output was read more specifically so was able to include p values up to <0.0001.

Reviewer 3 Report
Comments and Suggestions for Authors
Thank you for submission of your article. I found your findings compelling. Please have a look at my comments related to what I consider the main determinant(s) of the differences in outcomes based on previous epidemiological results and your current findings.
With highest respect

Author Response
Response: Thank you for recognizing the important step of this work towards the goal of trying to determine the factors of worse outcomes in Hispanic patients in California not only in thyroid cancer but in all cancers. We acknowledged some of your points regarding Hispanics having the lowest health insurance rate of any racial or ethnic group including more than twice of that of Non-Hispanic Whites and having limitations to financial resources and access of care especially during the time of the ACA and have expanded upon this in lines 929-935. We also agree with your point about acculturation as was noted in lines 907-909. Finally, we also acknowledge your point about a certain percentage of Hispanics being undocumented and not being identifiable as a result; this can also underestimate the magnitude of the disparity that Hispanics face and we have added a couple sentences noting this in our discussion in lines 1047-1050.

Reviewer 4 Report
Comments and Suggestions for Authors
This is an analysis of thyroid cancer in the Hispanic population from the California Cancer Registry. It is an interesting article, and it is coherent. However, before accepting it, the authors should consider some major and minor points, as referred to below:
MAJOR CONSIDERATIONS:
Papillary thyroid carcinoma tends to spread early via the lymphatic route to regional lymph nodes. On the contrary, follicular thyroid carcinoma does not metastasize to lymph nodes and when it metastasizes, it does so via the hematogenous (venous) route to different organs. It is known that in papillary thyroid carcinoma, lymph node metastases are associated with greater morbidity, but are not associated with mortality. In contrast, venous invasion is associated with mortality in both papillary thyroid carcinoma and follicular thyroid carcinoma. For this reason, both the 8th edition of the American Joint Committee on Cancer (AJCC) staging (online version) and the protocols of the American College of Pathology indicate that lymphovascular invasion should not be used in differentiated thyroid cancer and venous invasion should be specified. In papillary thyroid carcinoma, lymphatic invasion correlates with lymph node metastasis. Given that in general and in the present study most thyroid carcinomas are papillary thyroid carcinomas, the inclusion of the parameter “lymphovascular invasion” represents a confounding factor. The authors should eliminate references to lymphovascular invasion from the entire text. The authors should record only “venous invasion” in the study and add pertinent comments in this regard.
In the sixth paragraph of the discussion, the authors make a too direct statement between tumor size and the BRAF V600E mutation. This approach is too simplistic, in fact, some studies (PMID: 34885148; 32827026) have found a relationship between the BRAF V600E mutation and smaller size of papillary thyroid carcinomas. The authors should present the possible relationships in a more realistic context and consider the complexity of the genetic factors that may be involved.
MINOR CONSIDERATIONS:
There are typographical errors throughout the text. Authors should remove hyphens that appear within words throughout the article.
Author Response
Response: Thank you for your comments. Noted, with regards to the lymphovascular invasion data, we removed it from our analysis, though this did not change the overall findings of the results. For venous invasion, we only have 56 total cases of thyroid cancer with “venous invasion” only in our data set and cannot separate cases that have both lymphovascular invasion and venous invasion so consequently we decided to not include this as part of the analysis. With regards to BRAF V600E, the aim of including this information was to bring a push towards future studies to better explain these persistent differences in mortality risk and gender imbalance in Hispanic thyroid cancer cases in California, also in the limited studies looking at Hispanics and BRAF V600E, tumor size was the major finding. That being said, this was small sample size and needs additional studies for validation. Therefore, we have made some changes in the discussion to soften this stance and to call for additional studies to better elucidate the possible role of BRAF V600E in thyroid cancer and race/ethnicity (lines 930-958).

Round 2
Reviewer 1 Report
Comments and Suggestions for Authors
I still stand behind my original comments. However, the authors made some effort to improve the manuscript and adopted some of the suggestions, therefore I am willing to recommend the manuscript for publication.
Reviewer 4 Report
Comments and Suggestions for Authors
The authors have responded positively to my comments. In my opinion, the article is now appropriate for publication.